# Mammalian and Avian Larval Schistosomatids in Bangladesh: Molecular Characterization, Epidemiology, Molluscan Vectors, and Occurrence of Human Cercarial Dermatitis

**DOI:** 10.3390/pathogens11101213

**Published:** 2022-10-20

**Authors:** Sharmin Shahid Labony, Md. Shahadat Hossain, Takeshi Hatta, Anita Rani Dey, Uday Kumar Mohanta, Ausraful Islam, Md. Shahiduzzaman, Muhammad Mehedi Hasan, Md. Abdul Alim, Naotoshi Tsuji

**Affiliations:** 1Department of Parasitology, Faculty of Veterinary Science, Bangladesh Agricultural University, Mymensingh 2202, Bangladesh; 2Department of Parasitology and Tropical Medicine, Kitasato University School of Medicine, 1-15-1 Kitasato, Minami, Sagamihara 252-0374, Kanagawa, Japan; 3Department of Parasitology and Microbiology, Sher-e-Bangla Agricultural University, Sher-e-Bangla Nagar, Dhaka 1207, Bangladesh; 4Department of Fisheries Technology, Faculty of Fisheries, Bangladesh Agricultural University, Mymensingh 2202, Bangladesh

**Keywords:** schistosomiasis, swimmers’ itch, snails, human cercarial dermatitis, *Trichobilharzia szidati*

## Abstract

Schistosomiasis is a neglected tropical disease (NTD) caused by blood flukes (*Schistosoma* spp.). Schistosomatids affect a wide array of vertebrate hosts, including humans. In the present study, multiple species of schistosomatids were identified by isolating schistosomatid cercariae (SC) from naturally infected snails. We also described different biotic and abiotic factors influencing SC infections in snails and reported human cercarial dermatitis (HCD) for the first time in Bangladesh. A total of 22,012 snails of seven species: *Lymnaea auricularia*, *L*. *luteola*, *Indoplanorbis exustus*, *Physa acuta*, *Viviparus bengalensis*, *Brotia* spp., and *Thiara* spp., were collected and examined. Among these snails, 581 (2.6%) belonging to five species: *L*. *luteola*, *L*. *auricularia*, *P*. *acuta*, *I*. *exustus*, and *V*. *bengalensis*, were infected with SC. The rate of infection was the highest for *L*. *luteola* (11.1%), followed by *L*. *auricularia* (5.3%), and was the lowest for *V*. *bengalensis* (0.4%). Prevalence in snails was the highest in September (16.8%), followed by October (9.5%) and November (8.8%), and was the lowest in colder months, such as January (1.8%) and February (2.1%). Infections with schistosomatids were more common in larger snails and snails collected from sunny areas. We confirmed the presence of *Schistosoma indicum*, *S*. *incognitum*, *S*. *nasale*, *S*. *spindale*, and *Trichobilharzia szidati* by PCR and sequencing. Through a questionnaire survey, we detected HCD in 214 (53.5%) individuals, and the infection rate was almost equally distributed across all professions. Collectively, the present results suggest that lymnaeid snails are the main vector for *Schistosoma* spp. prevalent in Bangladesh, and schistosomatids with zoonotic potential are also prevalent.

## 1. Introduction

Schistosomiasis is a neglected tropical disease (NTD) caused by blood flukes (Trematoda: Schistosomatidae), and many schistosomatids have a zoonotic impact. Over 250 million individuals and millions of domestic animals and birds worldwide have active infections [1]. These parasites have been estimated to infect 700 million cattle worldwide, with more than 165 million cattle being affected in the regions of Africa and Asia, resulting in immense economic losses [1,2,3,4,5]. Globally, schistosomiasis is the fourth most prevalent helminth infection in humans, and 779 million individuals are at risk to the infection. Schistosomiasis is more debilitating than leishmaniasis, and accounts for 3.31 million disability-adjusted life years (DALYs) per year [1,6]. The disease is endemic to 78 subtropical and tropical countries throughout Asia, Africa, and South America, causing an estimated 280,000 deaths annually. *Schistosoma haematobium* and *S*. *mansoni* mainly infect humans, whereas *S*. *japonicum* is a zoonotic pathogen [7]. On the other hand, *S*. *bovis*, *S*. *indicum*, *S*. *spindale*, and ~40 species of *Trichobilharzia* affect animals and birds [8,9]. An adult schistosomatid may survive for 3–10 years within vertebrate hosts and may lay 300 eggs per day [7]. Following transdermal infection, cercariae transform into schistosomulae and travel to the portal veins of the liver, where they mature and pair [10]. In pairs, they reach their predilection sites, such as mesenteric veins (*S*. *bovis*, *S*. *indicum*, *S*. *japonicum*, *S*. *mansoni*, and *S*. *spindale*), urinary bladder veins (*S*. *haematobium*), and veins of the nasal mucosa (*S*. *nasale*, *T*. *arcuata*, *T*. *australis*, and *T*. *regent*), and lay eggs [8,9]. Eggs stick to the liver, urinary bladder, and nasal mucosa and induce granulomas characterized by the infiltration of lymphocytes, eosinophils, and activated macrophages [1,10]. In addition, the cercariae of some non-human schistosomatids, particularly those of avian schistosomatids, induce a skin reaction in humans that is characterized by severe urticaria and pruritus, which is known as swimmers’ itch or human cercarial dermatitis (HCD) [8].

Similar to most digenetic trematodes, schistosomatids require a snail intermediate host to complete their life cycle [9,11]. Since schistosomatids affect a wide array of vertebrates as definitive hosts, it is difficult to isolate and identify different species of schistosomatids from each species of definitive host. However, compared to the diversity of species of vertebrate hosts, fewer species of freshwater snails (FWS) act as vectors for schistosomatids in a particular area. In this context, a few studies have been conducted to detect various species of schistosomatids by collecting schistosomatid cercariae (SC) from naturally infected snails in different countries [12,13,14] and monitoring cercariae in water bodies [15,16]. Copro-PCR has been used to diagnose human schistosome infections in patients living in endemic areas [17]. We, herein, identified multiple species of schistosomatids that infect animals and birds (some with zoonotic potential) directly from naturally infected snails using PCR and sequencing, and identified the vectors of schistosomatids along with several factors affecting infections with SC in FWS. We also reported the prevalence of HCD for the first time in Bangladesh.

## 2. Materials and Methods

### 2.1. Ethical Approval

Research was conducted with approval from the local ethics committee (approval no: AWEC/BAU/2019 [60]). We did not handle any human or vertebrate samples/tissues. No vertebrate animals were killed or injured during this study. We interviewed participants who signed written consent forms.

### 2.2. Study Area, Sample Size, and Collection of Snails

The present study was conducted between 2019 and 2021 in different areas of Mymensingh, which is crisscrossed by many rivers, canals, and low-lying areas or wetlands (Figure 1). We collected and examined a total of 22,012 snails of seven species, namely, *Lymnaea luteola*, Lamarck, 1822 (Gastropoda: Lymnaeidae) (*n* = 3240), *Lymnaea auricularia*, Linnaeus, 1758 (Gastropoda: Lymnaeidae) (*n* = 3102), *Indoplanorbis exustus*, Deshayes, 1833 (Gastropoda: Planorbidae) (*n* = 3002), *Physa acuta*, Draparnaud, 1805 (Gastropoda: Physidae) (*n* = 2980), *Viviparus bengalensis*, Lamarck, 1822 (Gastropoda: Viviparidae) (*n* = 3140), *Brotia*, Adams, 1866 spp. (Gastropoda: Thiaridae) (*n* = 3200), and *Thiara*, Röding, 1798 spp. (Gastropoda: Thiaridae) (*n* = 3348).

Snails were manually collected from various water bodies using scoop nets. After collection, snails were gently washed with fresh water and snail species were identified based on morphological features [18]. During collection, snail habitats (i.e., the type of water body, micro-habitats in water bodies, and the presence/absence of direct sunlight) were recorded. 

### 2.3. Shedding and Tentative Identification of SC by Morphological Features

Snails were kept in separate glass test tubes containing water, and the shedding of cercariae was induced by exposing snails to sunlight or a lighting source that mimicked sunlight (550 Lumen) for 3 h as per previously reported procedures [11]. After snails shed cercariae, a drop of water containing cercariae was placed on a glass slide and examined under a microscope by adding 2% iodine solution. Snails shedding SC (furcocercous cercariae; cercariae with a head and bifurcated tail) were selected and recorded. Snails that did not shed cercariae were crushed, and the hemolymph was examined.

### 2.4. DNA Extraction

We extracted DNA from a pool of SC of each month (if any) from each snail species and processed it separately. Briefly, SC recovered from each species of snails was taken in 14 ml Falcon tubes separately and incubated on ice for 30 min. Then, the samples were centrifuged at 5000 rpm for 2 min. The supernatant was discarded, and the pellet was re-suspended by adding 500 µL of absolute ethanol in each tube and kept at −20 °C until further use. During use, cercariae recovered from each snail species were pooled and kept separately to make a FWS-wise stock. A pool of ~10,000 SC from each stock was then subjected to DNA extraction. Before DNA extraction, pellets were washed extensively with ice-cold PBS for 3 times to remove ethanol. Genomic DNA was extracted using the QIAmp DNA Mini kit (Qiagen, Hilden, Germany) following the manufacturer’s instructions. The concentration of DNA was estimated and then stored at −20 °C awaiting for further use. 

### 2.5. PCR and Agarose Electrophoresis

To identify the species of mammalian schistosomatids, PCR was conducted targeting 28S rRNA genes. During PCR, a fragment of the 28S rRNA gene was amplified using a primer set (SS 28, forward: 5′-CTG AGC TAC CCT TGG AGT CG-3′ and reverse: 5′-CAC TGA CAA GCA GAC CCT CA-3′) following previously reported procedures [19]. Briefly, PCR was conducted in a final volume of 25 µL using the One Taq Quick-Load 2X Master Mix (New England BioLabs Inc., Ipswich, MA, USA), 10 pmol of each PCR primer, and 50 ng of genomic DNA. The PCR thermocycling profile comprised an initial denaturation at 94 °C for 5 min, followed by 35 cycles of denaturation (94 °C, 1 min), annealing (60 °C, 30 s), extension (72 °C, 1 min), and a final extension step at 72 °C for 7 min. To identify avian schistosomes, we amplified the cox1 gene using previously reported primers (CO1F15: 5′-TTT NTY TCT TTR GAT CAT AAG C-3′ and CO1R15: 5′-TGA GCW AYH ACAAAY CAH GTA TC-3′) [20]. A PCR mixture (25 µL) containing water instead of template DNA was used as a negative control. Genomic DNA extracted from adult *Schistosoma* spp. isolated from bovine mesentery was used as a positive control. PCR products were subjected to agarose electrophoresis to detect transcripts of the expected sizes following previously reported methods [21].

### 2.6. Sequencing

After PCR, amplicons were purified by gel extraction using a NucleoSpin Gel and PCR clean-up kit (Macherey−Nagel, Düren, Germany) and sequenced in both directions with the same primers for 28S rRNA, and using the internal sequencing primer (CO1RH3R: 5′-TAA ACC TCAGGA TGC CCA AAA AA-3′) for cox1 by employing the BigDye Terminator v3.1 Cycle Sequencing Kit (Applied Biosystems, Waltham, MA, USA) on the ABI3500 Genetic Analyzer (Applied Biosystems). The recovered sequences were assembled using BioEdit 7.2 (https://bioedit.software.informer.com/7.2/; accessed on 10 March 2022) software and deposited to GenBank under the accession numbers ON597438-ON597447 and ON597471-ON597474.

### 2.7. Genetic Data Analysis and Phylogenetic Study

Identical sequences of schistosomatids were searched using BLAST^®^ (https://blast.ncbi.nlm.nih.gov/Blast.cgi, accessed on 10 March 2022), and sequences with higher identity (>99%) were selected and collected (DDBJ/EMBL/GenBank accession numbers are KF425713, KF425714, JQ408705, KP734296, FJ174496, JF838202, JF838203, AY157189, MT708489, and MT708493) [22,23,24,25,26,27]. The sequences of our isolates were aligned using CLUSTAL Ω (https://www.ebi.ac.uk/Tools/msa/clustalo/; accessed on 10 March 2022) for pair-wise comparisons with previously published sequences. Phylogenetic trees were constructed using maximum likelihood (ML), neighbor-joining (NJ), and minimum parsimony (MP) methods in MEGA X [23], based on the Tamura-Nei model with bootstrapping (1000 replicates) and otherwise default settings [28]. A 50% cut-off value was implemented for the consensus tree, and the 28S rRNA (AY 157239) or cox1 (AY 157185) gene of *Chemaerohemecus trondheinensis* (Trematoda: Strigeidae) was used as an outgroup.

### 2.8. Survey of HCD and Case Definition

A study was conducted using a structured questionnaire among individuals living in the river basin or around wetlands. Four hundred individuals participated in this survey, and the purpose of the study was clearly described to the participants in a language in which they were fully conversant. In a written consent letter, we confirmed that confidentiality about the personal information of participants would be strictly maintained, and the personal information of participants would not be handed over to a third party. We collected information only from individuals who willingly signed the consent letter. A history of HCD was revealed via interviewing the participants and close inspections. Itching that started at least 30 min after contact with natural water bodies was considered as to be HCD. Individuals with a history of pre-existing skin diseases were requested not to participate in the study. Participants willingly described their experiences. Photographs were taken only after the prior permission of participants. Since most of the participants in the present study were engaged in multiple professions, we constructed a Venn diagram using Bio-Venn software (https://www.biovenn.nl/index.php; accessed on 20 May 2022) to show the distribution of conditions in different professions. 

### 2.9. Statistical Analysis

To estimate the level of significance of SC prevalence in various species of snails and to demonstrate spatial and temporal distributions of vectors, we conducted statistical analyses using z-tests by employing SPSS software, and *p* < 0.05 was considered to be significant. Here, we used the z-test because our sample size is >30, our data was randomly selected from a population, and each item had an equal chance of being selected [29].

## 3. Results

### 3.1. Vector Snails of Schistosomatids Prevalent in Bangladesh

A comparable number of snails of different species, namely, *L*. *luteola* (*n* = 3240), *L. auricularia* (*n* = 3102), *I*. *exustus* (*n* = 3002), *P*. *acuta* (*n* = 2980), *V*. *bengalensis* (*n* = 3140), *Brotia* spp. (*n* = 3200), and *Thiara* spp. (*n* = 3348) were collected and examined. Schistosomatid cercariae were isolated from 581 (2.64%) snails belonging to five species: *I*. *exustus*, *L*. *auricularia*, *L*. *luteola*, *P*. *acuta*, and *V*. *bengalensis* (Figure 2A). Infections with schistosomatids were not detected in *Brotia* spp. or *Thiara* spp. (Figure 2B). In an initial survey, SC were identified by observing their characteristic morphological features. Each SC (furcocercous cercariae) was characterized by the presence of a head and bifurcated tail, resembling a tuning fork, which is known as a furca (Figure 2C).

### 3.2. Relative Infection Rates of SC in Different Snail Species in Bangladesh

We collected a comparable number of snails of each species. The infection rate with SC, the pool for all schistosomatid species involved, was significantly (*p* < 0.05) higher in *L*. *luteola* (11.1%) and *L*. *auricularia* (5.3%) than all other snail species. The lowest prevalence was detected in *V*. *bengalensis* (0.4%) (Figure 2B). We observed that infection rates were seemingly higher in bigger snails than in smaller snails. *Lymnaea auricularia*, *L. luteola*, and *I. exustus*, which were bigger than 16 (20 ± 5.5), 9 (12 ± 6.25), and 7 (9.7 ± 5.1) mm, respectively, had a higher rate of infections. Very small snails (in the case of lymnaeid snails < 7 mm and planorbid snails < 5 mm in size) did not carry SC.

### 3.3. Effects of Seasons on the Infection of Snails with Schistosomatids

A similar number of snails were collected in January (*n* = 1850), February (*n* = 1840), March (*n* = 1802), April (*n* = 1840), May (*n* = 1801), June (*n* = 1817), July (*n* = 1900), August (*n* = 1850), September (*n* = 1812), October (*n* = 1805), November (*n* = 1888), and December (*n* = 1817). During the study period, we detected SC in the vector snails of various species more or less throughout the year. However, between September and November, the recovery rate of SC was the highest. The rate of infection was the highest in September (16.8%), followed by October (9.5%) and November (8.8%). In contrast, the rate of infection was low in January (1.8%), February (2.1%), and April (1.4%). The prevalence of SC was also lower in hotter months, such as June-August (Figure 3A).

### 3.4. Different Habitats of Vector Snails of Schistosomatids

A comparable number of snails were collected from different water bodies, such as a river (*n* = 4412), canals (*n* = 4399), beels (*n* = 4418), paddy fields (*n* = 4401), and drains (*n* = 4382) and habitat-wise infection rates were estimated. Among the different habitats examined, the highest percentage of SC was recovered from snails collected from paddy fields (7.9%) and the lowest percentage (0.4%) from those in drains (Figure 3B). We recorded sunlight exposure when collecting snails. Snails collected from a shaded place were less likely to be infected with SC (2.5%) than those from sunny areas (10.5%), suggesting that infected snails possibly prefer the sunny areas of water bodies. 

### 3.5. Confirmation of Species of Schistosomatids by PCR and Sequencing

Since all species of schistosomatids generate furcocercous cercariae with similar morphology then to confirm different species, we used molecular tools. By PCR, we obtained amplicons at the expected level. PCR products were sequenced and analyzed. In a BLAST search, our newly recovered sequences of Bangladeshi isolates (ON597438-ON597447 and ON597471-ON597474) were mostly identical (99.41–100% identities) to the previously reported sequences of *S*. *incognitum*, *S*. *indicum*, *S*. *nasale*, *S*. *spindale*, and *T*. *szidati*; thus, we unambiguously identified the species. However, pair-wise analysis of our newly recovered sequences with the reference sequences using CLUSTALΩ revealed few (1–3) single nucleotide polymorphisms (SNPs), which may be due to a change of geographical location. Our sequencing effort did not detect any matches with sequences specific for *S. mansoni*, *S. japonicum*, and *S. haematobium*, suggesting that these primary human schistosomes were not prevalent in Bangladesh, at least in our study areas. We then built a phylogram with our newly recovered sequences of the 28S rRNA gene from our isolates using the reference genes of *S*. *haematobium*, *S*. *indicum*, *S*. *incognitum*, *S*. *japonicum*, *S*. *mansoni*, *S*. *nasale*, and *S. spindale*. The 28S rRNA gene sequences assigned under GenBank accession numbers ON597438-ON597440, ON597441-ON597443, ON597444-ON597446, and ON597447 formed distinct clusters with the reference genes of *S*. *indicum* (accession number: KF425714), *S*. *nasale* (accession number: KP734296) [27], *S*. *spindale* (accession number: KF425713), and *S*. *incognitum* (accession number: JQ408705) [30], respectively, with strong nodal support (Figure 4A), thereby providing further validation of the species of schistosomatids prevalent in Bangladesh. We built another phylogram using new cox1 sequences of our isolates and those of *Austrobilharzia* sp., *Gigantobilharzia melanoidis*, *Ornithobilharzia canaliculata*, *T*. *anseri*, *T*. *franki*, *T*. *mergi*, *T*. *ocellata*, *T*. *physellae*, *T*. *querquedulae*, *T*. *stagnicolae*, and *T. szidati.* Similarly, our cox1 sequences (GenBank accession numbers: ON597471-ON597474) formed a cluster with *T*. *szidati* (accession numbers: FJ174496, JF838202, JF838203, MT708489, and MT 708493), supported by strong bootstrap values. However, our isolates were distantly related to the sequences of *T*. *mergi* (JX456172), *T*. *stagnicolae* (FJ174493), *T*. *physellae* (FJ174522), *T*. *querquedulae* (FJ174500), *T*. *franki* (HN131200), *T. anseri* (KP901384), *G*. *melanoidis* (JX875069), and *O*. *canaliculata* (AY157194) (Figure 4B), further confirming the species as *T*. *szidati.* By molecular analysis, we revealed that the SC of *T*. *szidati* were more common in *P*. *acuata* than in lymnaeid snails. We also got avian SC in lymnaeid snails and *V. bengalensis* as well, and *V. bengalensis* harbored only *T. szidati*. On the other hand, *S. indicum*, *S. spindale*, *S. nasale*, and *S. incognitum* mainly infected lymnaeid and planorbid snails. None of the vertebrate schistosomatids, identified in this study, were isolated from *V*. *bengalensis*. 

### 3.6. Distribution of HCD across the Different Professions in the Study Areas

Since we confirmed the presence of *T. szidati*, which is an avian schistosome that has been reported to cause HCD [31], we surveyed HCD in the study areas by interviewing individuals engaged in farming and/or fishing as an occupation or those who used natural water bodies for swimming or bathing. The survey results showed that 214 (53.5%) individuals were affected by HCD. The condition was almost equally distributed among individuals irrespective of their profession (Figure 5A). Participants in the survey had the experience of being affected by HCD while working in paddy fields, fishing, or swimming and bathing. Some farmers showed us the affected areas of their legs, and lesions were characterized by the presence of urticaria and intense pruritus (Figure 5B). However, in old cases, pruritus subsided and urticarial lesions became blackish (Figure 5B, right panel). Participants mainly worked in shallow water for paddy farming, a common scavenging area for ducks and other aquatic birds. Non-human cercariae, particularly those of avian origin, possibly released by FWS and penetrate the skin, resulting in HCD (Figure 6).

## 4. Discussion

Schistosomatids essentially propagate asexually (parthenogenesis) within FWS and release cercariae, the freely swimming infective stage for vertebrate hosts; therefore, the diseases caused by the blood flukes are one-health or eco-health related issues. In the present study, we confirmed the prevalence of five species of schistosomatids such as *S*. *indicum*, *S*. *incognitum*, *S*. *nasale*, *S*. *spindale*, and *T*. *szidati* in Bangladesh. We also identified the different molluscan vectors required for the propagation of schistosomatids in Bangladesh, along with different biotic and abiotic factors that govern infections in snails with SC. Furthermore, we documented HCD among a population living in the study areas.

The present results revealed that *I*. *exustus*, *L*. *auricularia*, *L*. *luteola*, *P*. *acuta*, and *V*. *bengalensis* acted as the vectors of schistosomatids in the study areas of Bangladesh. In a previous study, *L. auricularia* and *L. luteola* were identified as the vectors of *Schistosoma* spp. in Bangladesh [32]. We found the highest infection rate in lymnaeid snails, indicating that they may be the main vectors of schistosomatids in the study areas in Bangladesh. Among lymnaeid snails, higher infections were found in *L. luteola*, suggesting that this snail plays a substantial role in the existence of schistosome parasites in Bangladesh. In the present study, the recovery rate of SC from *I*. *exustus* was also slightly higher than that from the other species (except lymneaid snails) of snails examined and identified as vectors of schistosomatids. *Indoplanorbis exustus* is a planorbid snail and FWS belonging to the family Panorbidae play a crucial role in the transmission cycle of human schistosomes in different countries [33]. We also found schistosomatid infections from *P*. *acuta*, an important vector of avian schistosomes [34]. Adult avian schistosomes had already been isolated from ducks in Bangladesh [35]. Although there is currently no information on HCD in Bangladesh, field experience suggests that swimmers’ itch is also prevalent in Bangladesh, which s mainly caused by the penetration of human skin by non-human SC, particularly by the cercariae of avian schistosomes [8]. The present results revealed avian SC in *V*. *bengalensis*, which were collected from a free-flowing drain. Notably, these drains are common scavenging areas of domestic ducks. Lymnaeid and planorbid snails, the main vectors of schistosomatids, are not typically found in drains; therefore, *Viviparus* snails may act as surrogative vectors in the absence of the main vectors. However, there was previously no evidence to suggest that *V*. *bengalensis* acts as a vector of any species of schistosomatids. To our knowledge, this is the first study to confirm *V*. *bengalensis* as a vector of schistosomatids. 

The present study revealed that the seasons of the year had a marked impact on the infection rates of snails by SC. We found the highest infection rates between September and November and the lowest in colder months, such as January and February, in Bangladesh. In the winter months, FWS hibernate and become less active. This phenomenon may play a vital role in the lower harvest of cercariae in the colder months. On the other hand, we found that a very few number of snails were infected with SC in hotter months, such as June and July, which is difficult to explain. However, in hot weather, snails also become inactive. The scorching heat of the hotter months may also have adverse effects on the development of schistosomatids in snails. We reported similar findings during the recovery of echinostome cercariae from FWS [36]. Bangladesh is a small country that is mainly situated in a tropical region and partly in subtropical regions. It has a very short winter (from December to February), during which temperatures may decrease to below 10 °C. In the summer (March to May) and even in the rainy season (June to August), temperatures may increase up to 40 °C. Since the activity of snails is minimal in the hot and cold months, the relatively lower prevalence of SC in these periods is pertinent. 

Snails collected from paddy fields and small free-flowing natural canals or rivers showed the highest infection rate. These types of natural water bodies are the principal habitats of lymnaeid snails, the main vector of schistosomatids in Bangladesh. Therefore, higher rate of recovery of SC from FWS collected from these habitats are also quite pertinent. After harvesting crops, paddy fields are used as pastures for cattle and buffaloes, the major definitive hosts of bovine schistosomes. During grassing, a large number of cattle and buffaloes may become infected at these pastures. On the other hand, infected animals also contaminate the land by shedding eggs with feces; therefore, paddy fields (as single habitats) markedly contribute to the completion of the life cycles of schistosomatids. There are about 1.5 million buffaloes in Bangladesh [37]. Naturally, buffaloes prefer muddy or marshy places to highland pastures. Buffaloes are fond of spending more time in low-lying areas for wallowing. Therefore, all excreta are mixed in water. In addition, paddy fields and natural canals are the main scavenging areas of domestic ducks and other wild aquatic birds. There are about 54 million domestic ducks in Bangladesh, which are mostly reared by the villagers in the scavenging system [38]. During the day time, they scavenge in the water bodies to collect food and continuously contaminate waterbodies with their feces; thus, they are the main source of contamination for avian schistosomatids. In Bangladesh, rivers and canals are commonly used for the bathing of livestock. 

Interestingly, the rate of infections was higher in snails collected from sunny areas. Exposure to sunlight or artificial light with the same intensity as sunlight is vital for aggressing cercariae from infected FWS [11]. The exposure of infected snails to light is commonly used to harvest cercariae [2]. Therefore, it is quite logical that infected snails may migrate to sunny areas to be exposed to the sunlight and release cercariae. 

By using PCR and subsequent sequencing, we identified *S*. *indicum*, *S*. *incognitum*, *S*. *nasale*, *S*. *spindale*, and *T*. *szidati*. In PCR, we used primer sets that amplified the 28S rRNA or cox1 gene of all schistosomatids and have been successfully used in previous studies [19,21,39]; thus, PCR and the subsequent visualization of amplicons at the expected level confirmed our tentative identification of SC. However, PCR using global primer sets specific for the family or genus cannot distinguish among species; therefore, we sequenced amplicons and then performed bioinformatic analysis. Our newly recovered sequences (ON597438-ON597446) showed very high identities (99–100%) to the previously reported reference sequences of *S*. *indicum*, *S*. *incognitum*, *S*. *nasale* and *S*. *spindale*, thereby confirming the presence of the species in Bangladesh. There were very high similarities in the sequences of the 28S rRNA genes of *S*. *incognitum*, *S*. *intercalatum*, and *S*. *spindale*; therefore, they formed a single cluster, but each of the species formed distinct sister clusters. *Schistosoma indicum* and *S. spindale* cause visceral schistosomiasis in sheep, goats, cattle, and buffaloes, resulting in fetid diarrhea [40]. The ova of visceral schistosomes are encountered in routine fecal examinations in Bangladesh. On the other hand, *S. nasale* causes nasal granuloma, resulting in snoring and dyspnea [41], and snoring disease is sporadically encountered in cattle in Bangladesh. It is important to note that our molecular studies failed to detect major human schistosomes, such as *S*. *mansoni*, *S*. *japonicum*, and *S*. *haematobium* in the study areas; however, we confirmed the presence of *S. incognitum*, which has zoonotic potential. *Schistosoma incognitum* has already been isolated from rats [42] in Bangladesh. Additionally, we confirmed one species of avian schistosome that is frequently linked to swimmers’ itch, commonly known as ‘pani kamor’ in Bangladesh. In a previous study, *T*. *szidati* was also detected in Bangladesh [31,43].

We revealed that a high percentage of participants (>50%) in the present study had a history of HCD, irrespective of their profession. Schisomatid cercariae generally accumulate near the margin of water bodies by wind, and shallow, stagnant water acts as a ‘hot spot’ of HCD; therefore, individuals working near the banks of water bodies are at high risk to the infection. Lymnaeid snails are the main vectors of mammalian and *Physa* spp. contribute to the life cycle of avian schistosomes, and both molluscan species are mainly present in paddy fields, rivers, and canals in Bangladesh. Therefore, individuals using natural water bodies or engaging in farming or fishing are generally affected by HCD. The present study was conducted among individuals living in the river basin or near wetlands. They engaged in multiple professions, which are shown in the Venn diagram. Individuals living in this area, irrespective of their profession, depend largely on natural water bodies for various purposes, such as bathing, washing clothes and crockery, swimming, fishing, and farming. Therefore, they are in continuous contact with sources of cercarial infection, which may make an important contribution to the high prevalence and indiscriminate distribution of HCD among individuals who had frequent exposure to water bodies harboring snails. HCD is a type I hypersensitivity reaction. When the cercariae of avian schistosomes penetrate human skin, they die and release several chemical mediators, resulting in allergic reactions. Activated mast cells and basophils release histamine, leading to vasodilation [8]. In the first exposure, the condition may remain unnoticed, whereas intense pruritus develops at the third, fourth, and repeated exposures [43]. In subsequent exposures, HCD may develop within 30 min of the penetration of cercariae, and pruritus persists for up to 8 days post-infection [8]. Due to persistent exposure, papules become blackish, and secondary bacterial infections may further aggravate the condition [44]. 

## 5. Conclusions

The present study confirmed that five FWS, *L. auricularia*, *L. luteola*, *I. exustus*, *P. acuta*, and *V.*
*bengalensis*, acted as the vectors of schistosomes in Bangladesh, of which lymnaeid snails were the main vectors. The species of FWS and their habitats, the size of snails, seasons, and sunlight influenced the prevalence of SC in molluscan vectors. Although human schistosomes were not prevalent in the study area, zoonotic schistosomes and the causal agents of swimmers’ itch were confirmed, suggesting that individuals in the country, particularly people living near low-lying marshy areas, are at risk.

## Figures and Tables

**Figure 1 pathogens-11-01213-f001:**
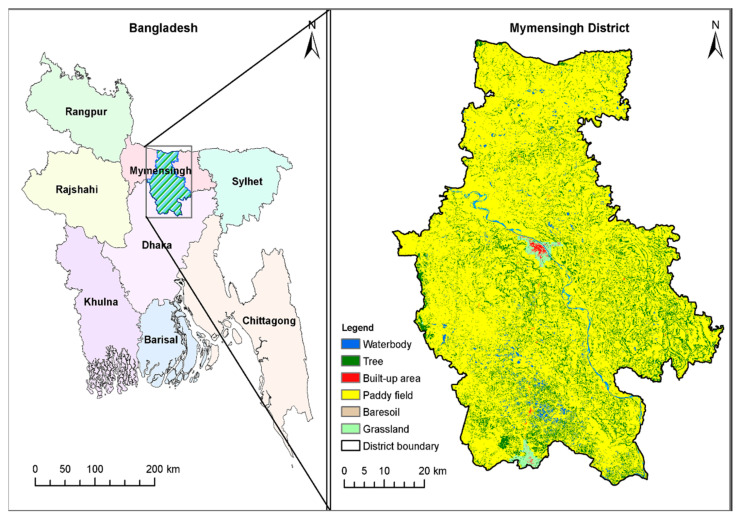
**Study area**. Prepared with Landsat satellite images and the Google Earth engine; see ref. [19] for detailed method.

**Figure 2 pathogens-11-01213-f002:**
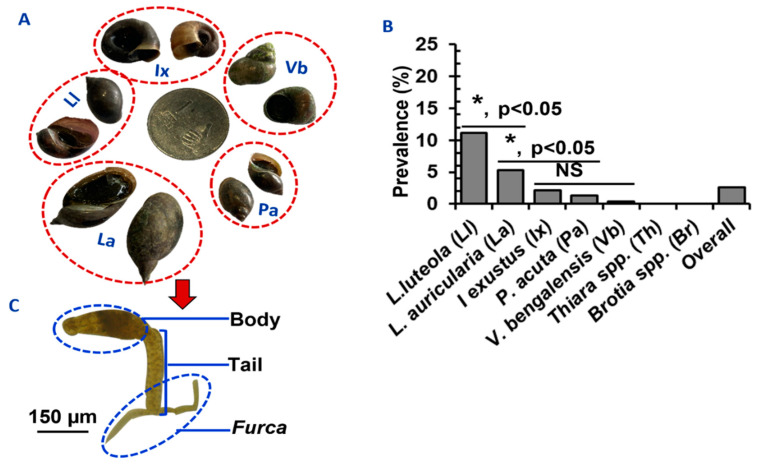
**Comparative prevalence of SC in different species of snails in Bangladesh.** (**A**) Species of the snails infected by *Schistosoma* spp. in Bangladesh. (**B**) Infection rates of SC in different snail species. * *p* < 0.05 were statistically significant. NS; statistically insignificant (*p* > 0.05). (**C**) A microphotograph of SC showing its characteristic features.

**Figure 3 pathogens-11-01213-f003:**
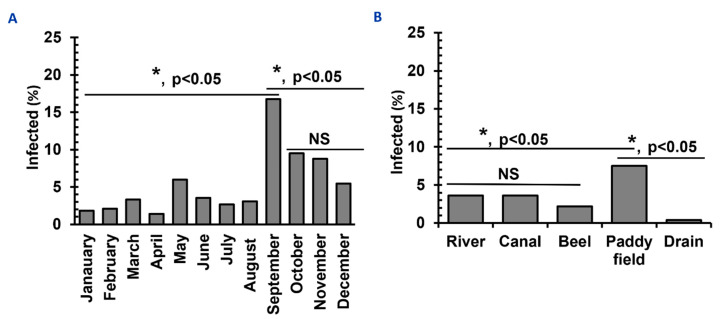
**Temporal distribution and habitat diversities of Schistosomatid-infected vector snails in Bangladesh.** (**A**) Temporal distribution of infections with schistosomatids in vector snails. (**B**) Habitat preference of vectors of SC. * *p* < 0.05 are statistically significant. NS, statistically insignificant (*p* > 0.05).

**Figure 4 pathogens-11-01213-f004:**
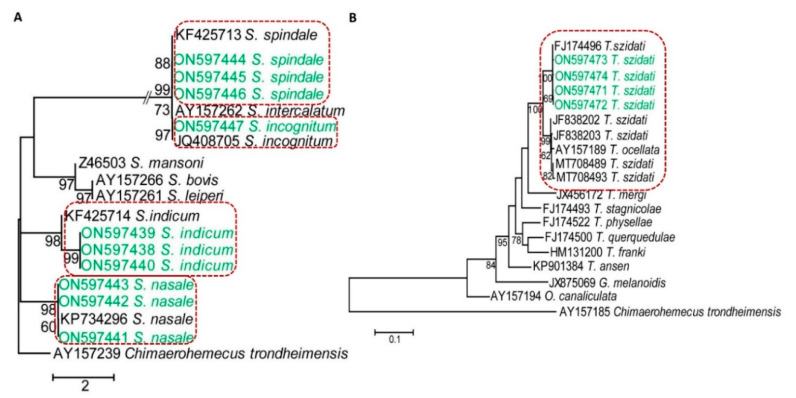
**Dendrograms showing species-specific clusters of sequences.** (**A**) A 28S rRNA-based dendrogram showing different species of schistosomatids affecting mammals. (**B**) A cox1 gene-based dendrogram of schistosomatids affecting birds. Phylogenetic trees were constructed using maximum likelihood (ML) methods in MEGA X.

**Figure 5 pathogens-11-01213-f005:**
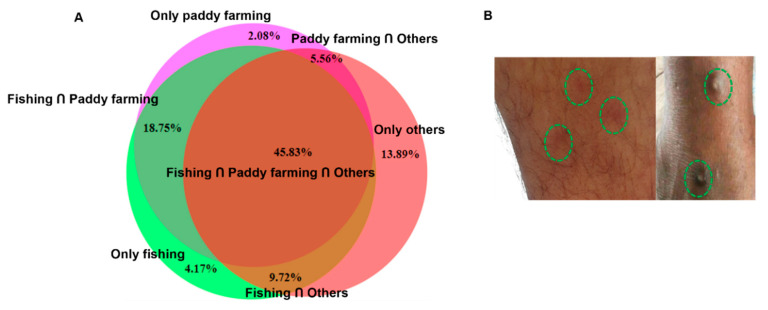
**Human cercarial dermatitis was highly prevalent in study areas in Bangladesh**. (**A**) Distributions of HCD across the different professions in the study areas were estimated by a questionnaire-based survey. (**B**) Skin lesions induced by HCD; left panel, acute HCD and right panel, chronic HCD.

**Figure 6 pathogens-11-01213-f006:**
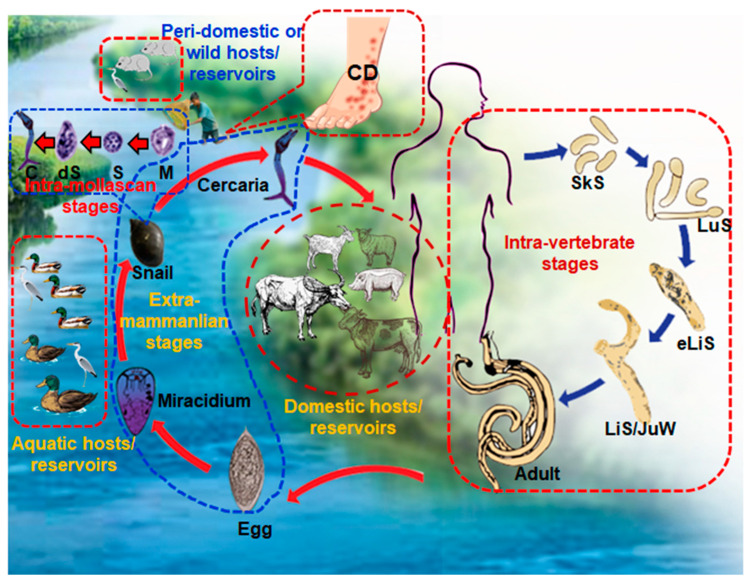
Schematic diagram showing the complex life cycle of *Schistosoma* spp. and a demonstration of exposure to cercariae leading to the development of human cercarial dermatitis (HCD). CD, cercarial dermatitis; SkS, skin stage; LuS, lung stage; eLiS, early liver stage; LiS/JuW, liver stage or juvenile worms; C, cercariae; dS, daughter sporocysts; S, sporocyst; M, miracidium.

## Data Availability

The data presented in this study can be found in the main text.

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
