# Peer review of "Mammalian and Avian Larval Schistosomatids in Bangladesh: Molecular Characterization, Epidemiology, Molluscan Vectors, and Occurrence of Human Cercarial Dermatitis"

_pathogens, 2022, doi:10.3390/pathogens11101213_

Round 1

Reviewer 1 Report

Dear authors

Overall, I find the core data being presented here novel and significant, and well done. These findings are worthy of publication. However, there are many issues with the manuscript and it cannot be published without major revision. Throughout, the level of English and grammar requires improvement. 

General issues

The structure of the manuscript is unusual:

1. Introduction

2. Results

3. Discussion 

4. Methods

5. Conclusion

Why not place Methods between Introduction and Results? The Results contains too much Methods-like explanation of how and why things were done; I suggest moving the Methods to before the Results will improve flow and allow the authors to avoid explaining methodologies in Results. Similarly, parts of the Discussion belong in Results, there is also quite a bit of background information that perhaps would be better placed in the Introduction.

It seems to me that nowhere do the authors make clear which species of schistosomatids infected which snails! Much of the discussion and analyses of results is undermined by the authors seemingly grouping all schistosomatids together - multiple species are involved!

I see no advantage of using the informal and ambiguous term "schistosome" over the unambiguous and correct term "schistosomatid", or "schistosomatoid" or "schistosoma spp." depending on the precise meaning intended in each context. 

Please check the names of snails used are valid. Several appear to listed as invalid names on MolluscaBase. Lymnaea luteola should be Racesina luteola (Lamarck, 1822). Lymnaeae auricularia should be Radix auricularia (Linnaeus, 1758). Physa acuta should be Physella acuta (Draparnaud, 1805). Vivipaara vivipara should be Viviparus viviparus (Linnaeus, 1758). Are the authors using an alternative taxonomy? I think the authors should consider providing authorities for the names at first use, and perhaps also higher taxa.

The authors make some unsubstantiated claims in their results section. Specifically, relating to "preference" for snails and prevalence related to light. The authors do not provide data nor explain their analyses. More importantly, the logic in both cases is seemingly flawed. See specific comments below.

The authors should provide references for the GenBank data used in their phylograms, to give due credit to the work of others.

Figure captions are inappropriate - the text mostly belongs in methods.

Species names must be italicised. Also, when abbreviating the genus, the period should not be italicised. Please check and correct throughout. 

Hyphens should only be used to hyphenate. Use en-dashes for ranges.

Lymnaeid is misspelled throughout.

Do not start sentences with abbreviations, including genera.

Specific corrections, queries and suggestions

Abstract

ln 17: "The disease" is a bit confusing here, because, in the previous sentence, schistosomiasis is described as a NTD, which, I presume only applies to diseases in humans, whereas here, the disease is being applied to broader taxa. When talking about such broad host taxa, and therefore broad schistosomatid taxa, or perhaps even schistosomatoid taxa (the authors use of "schistoome" makes the meaning here ambiguous), is it even accurate to talk about "the disease"? I think I would avoid all these difficulties by replacing "the disease affects" with "schistosomatids (or-oids, depending on the authors meaning here) exploit"

ln 17: Is the meaning of "animals" here "mammals"? Humans, birds, and indeed all vertebrates, are of course animals.

ln 18: I think here in particular it would be helpful to be precise: do the authors mean schistomsona spp., schistosomatids, or schistosomatoids?

ln 19: I think add "in Bangladesh" to the end of the sentence here.

ln 21-22: Move "were collected and examined" to after "namely", before the list of taxa. Taxa must be italicised. I'm not a malacologist, but according to MolluscaBase, several of the names used here are invalid: Lymnaea luteola should be Racesina luteola (Lamarck, 1822). Lymnaeae auricularia should be Radix auricularia (Linnaeus, 1758). Physa acuta should be Physella acuta (Draparnaud, 1805). Vivipaara vivipara should be Viviparus viviparus (Linnaeus, 1758).

ln 23: "snails", plural. Incorrect use of "such as", the authors list all five. Taxa must be italicised.

ln 25: "infections", plural.

ln 26: Replace "snail was the highest" with "snails were highest".

ln 27: Remove "in Bangladesh" - implicit by now.

ln 29-30: Italicise taxa. What is the order for the list here? Why not alphabetically?

ln 30: Hyphenate "cross-sectional" as it is a compound adjective modifying the noun.

ln 31: Wow, >50% seems like a lot! I presume a by "cross-sectional study" the authors mean it is representative of the general population. Pluralise "profession".

ln 32: Spelling of lymnaeid

Keywords: there is no point including words that appear in the title (perhaps even abstract?) in keywords - any search engine will consider the title anyway.

Introduction

ln 38: Why is schistosomiasis italicised?

ln 39: The correct term for members of the Schistomsomatidae is schistosomatids, not schistosomes.

ln 42: Insert "in" after "resulting". I'm not sure I understand why the following is included: "with more than 165 million cattle affected in the regions of Africa and Asia". Why not just leave the statement at 700 m cattle worldwide? Asia I suppose is relevant, because the focus is Bangladesh, but why Asia + Africa specifically?

ln 47: I think a reference is needed to support the deaths statistic here.

ln 48: Replace "where" with "whereas". "S. japonicum infects mainly humans whereas S. japonicum is a zoonotic pathogen" does not make sense.

ln 49: List taxa alphabetically?

ln 51: Use en-dash, not hyphen, for ranges. Replace slash with "per".

ln 53: Remove "up". Replace "In pair" with "Once paired".

ln 54: List taxa alphabetically?

ln 55: Insert "the" before "nasal".

ln 55-56: List taxa alphabetically?

ln 59: Insert comma after each "schistosomes" here.

ln 60: "Swimmers'" should be lowercase.

ln 62: This statement is harmless I suppose, but a bit misleading. Not all digeneans require a snail (some use polychaetes) and most digeneans actually require more than a snail, i.e. schistosomatoids are unusual in having a two vs three or four host life-cycle.

ln 65: Either remove comma after "Although" and remove "however," or simply start this sentence from "However,...". Use "varied" instead of "wide" here. Replace "a few" with "only few" or even just "few".

ln 66-67: Perhaps "several" vs "a few"?

ln 67: Replace "had" with "have", but, in any case, it is implicit that "studies" refers to studies that have been published, so simplify "studies have been published" to just "studies". Pluralise "parasite" becomes multiple species are involved. The sentence does not quite make sense, perhaps replace "studies had been published for the detection of the parasite in snails" with "studies reported detection of the parasites in snails".

ln 68: Remove "and".

ln 68-70: I don't understand the point being made by this sentence, nor why the sentence requires/justifies a "however"? Is the point that identification to species requires sequencing?

ln 71: Replace "potentials" with "potential".

ln 72: I think "identify" is better than "demonstrate"? And "intermediate hosts" or "molluscan hosts" is better than "vectors" because the snails are not really vectors (depending on definition) - they do not actively transmit or infect the pathogen to the vertebrate host.

ln 73: "affecting" or "predicting"? Delete "with the parasite".

Results

ln 76: Most of the text in this subsection does not belong in Results. I recommend using "host" vs "vector".

ln 77-78: This is simply not true. Some digeneans use bivalves, scaphopods or polychaetes as first intermediate hosts. Not all life cycles involve gastropods.

ln 77-79: This sentence seems more like Introduction than Results.

ln 79-82: This sentence reads more like Methods than Results, except perhaps for the number collected.

ln 81-82: Check species names are valid.

ln 83: Where is this section? Unusual for Results to come before Methods. Also unusual to state in Results that protocols were carried out as per the Methods section - of course that is the case, that is the whole point of a Methods section.

ln 83: "Fercocercus cercaria" should be lowercase and not italicised.

ln 84: Incorrect use of "such as". The authors mean "specifically" or "namely".

ln 84-85: When abbreviating Latin names, the period should not be italicised.

ln 86: An initial what? First clause does not make sense.

ln 86-89: This reads like Methods. Results should contain what was found. Methods is for how it was found. 

ln 100: I am concerned by the use of the word "preference" in this section. As far as I can tell, the authors are simply reporting prevalence of SC in five different species of snails. Some species had higher prevalence than others. The authors seemingly take this to mean schistosomatids have a preference for those species of snails? This logic is flawed, mainly because multiple species of schistosomatids are involved here. Presumably each species of schistosomatid involved has a specific snail or snails that it exploits. So the higher prevalence in some snails is likely linked to the relative rarity of each schistosomatid involved. Furthermore, for any snail that hosts more than one schistosomatid, the prevalence reported here is actually the sum of prevalences for those species. There is a second reason this logical is flawed: perhaps some snails are more common than others and so perhaps miracidia will infect a more common viable snail host if there preferred snail host is not available? The point is, to truly test preference, the authors would need to experimentally expose miracidia to a choice of snails. At this point in the manuscript, the authors have not yet made clear how many species of schistosomatids are involved, which species were found in which snails, nor whether any exploit more than one species of snail, nor whether any species of snail supports more than one schistosomatid life cycle. Or perhaps the "preference" referred to here is the size of the snails within each species? This is ambiguous and not clear. For size, the authors have provided no data or analyses to support their claim. Even if they had, the authors would need to be careful to disentangle or at least address two confounding factors before claiming miracidia prefer larger snails: first, some trematodes can cause gigantism in snails, second, larger snails are presumably older and thus more likely to have acquired an infection (i.e. there is no telling when a large snail was infected, perhaps it was first infected when it was a small snail).

ln 101-102: Again, this is more Methods (how and why it was done) vs Results (what was found). These problems could be resolved by placing the Methods before the Results. By now, it is established and implicit that all this work relates to "in Bangladesh"

ln 103-104: Remove the comma after "suggested that". The use of "suggested" is incorrect. The result is simply that L. luteola had the highest infection, it is incorrect to say the results suggested that L. luteola had the highest infection - it did have the highest infection.  

ln 104-105: I thought five snail species were infected, why are data for only three provided here?

ln 106: What is meant by "more" here? Greater frequency or greater intensity? And what is meant by larger vs smaller - presumably the authors mean within each snail species larger snails had "more" infections, not that larger snails of species had "more" infections than smaller species? The phrasing is ambiguous.

ln 107: These numbers would be more intuitive if the context of the range of sizes examined was provided.

ln 108: Quantify "higher rate". How much higher? Can the authors provide a function for likelihood of infection determined by size? Is the trend statistically significant?

ln 109: "Data" should be lowercase. Why are data not shown? If the authors have reasons to withhold these data, then fine, but they can't simultaneously claim that there is a trend between size and prevalence without providing any data or analyses. Either include the result + the data and perform some basic statistical analyses, or remove the unsubstantiated claims.

ln 114: "months" plural. "Frequency" is better than "rate" here.

ln 116: "lowest"

ln 118: Replace hyphen with en-dash.

ln 132: This subsection discusses "niches", but I think "habitats" would be better. A "river" is a bit broad to be considered a "niche".

ln 140-143: Is the light/shade confounded with habitat type? I.e. highest was paddies, lowest was drains, presumably paddies are high light environment and drains low light? But perhaps higher prevalence in paddies is due to relatively static water and proximity and density of vertebrate hosts, whereas drains perhaps have too much flow - i.e. nothing to do with light necessarily? Or did the analyses consider both factors? No explanation of analyses is provided. Again, the authors choose not to show their data/analyses and so make unsupported and opaque claims.

ln 145-150: This is all methods, not results.

ln 145-146: Either "almost identical" or "similar" but not "almost similar".

ln 146: Remove "therefore". Cannot use "therefore" together with "since" in this way.

ln 147: SC mentioned twice in sentence, once too many!

ln 147: By "confirm", the authors mean "identify"?

ln 149: Reference to methods would not be required if methods cam before results.

ln 149-150: I do not really understand the point of this sentence. What "expected level"? The authors mean the PCR was successful? I think not needed - implicit.

ln 151-152: Use en-dashes, not hyphens. Which genetic marker is being used here?

ln 154: The use of "retrieve" is confusing here. It reads like the authors were unable to procure (retrieve) sequences of these species from GenBank, but I think the intended meaning here is that none of their sequences matched those species?

ln 155-156: Well, I suppose so. However, the authors have not provided an indication of how many sequences they generated (they provide it later in the paragraph). Presumably they generated sequences from each species of snail infected? This is not made clear.

ln 157: Retrieved is the wrong word again here. Perhaps "generated" or "recovered"?

ln 158-174: This seems overly cumbersome. The point here is too identify species. I think, in addition to providing the figures, all that needs to be said is that the new sequences were identical to previous sequences and so unambiguously identified taxa. Except, I think, H. indicum? The new sequence differs slightly? The authors should address this and justify why they consider it the same species.

ln 189-190: Cannot use "since" with "therefore".

ln 180: Why "planned to survey"? Why not just "surveyed"?

ln 192: I'm not "amusement" is the best word here. Depending on whether the authors intend this to relate to fishing or farming or both, perhaps "recreational" or "artisanal" is better?

ln 201-203: This sentence is grammatically flawed.

Discussion:

ln 231-241: This paragraph is just a partial repeat of the background and a partial summary of the results. No discussion of results here.

ln 233: Genus needs to be italicised.

ln 233: Why "essentially"? They always propagate asexually.

ln 234: Is it known that this is actually parthenogenesis? Does the method of reproduction here qualify as parthenogenesis?

ln 234: Delete "to survive and exist in the nature".

ln 236: I do not understand "one health or eco-health related problem."

ln 237: Cercariae were not collected with molecular tools! And I think "several" is an exaggeration.

ln 242: Italicise snails.

ln 243: Present tense: "act".

ln 245-249: As in results, the logic is flawed here. Multiple schistosomatids are involved, perhaps some are more common than others and thus their specific snails have higher prevalence. The authors have still not yet revealed which species are in which snails.

ln 253: Cercariae cannot be recovered from a genus, only species of a genus.

ln 265: Again, a genus cannot be a host, only species of a genus.

ln 273: Remove "a bit" and "exactly".

ln 276: Improve grammar.

ln 289: This is not "on the other hand" (i.e. conversely), use "simultaneously".

ln 299-303: None of this is needed here. It is all methods.

ln 304-306: I am confused. In the results the authors identify their schistosomatids to species based on sequences. Here they reveal the markers cannot discriminate at the species level?

ln 307-316: None of this belongs here, this is all results, not discussion/interpretation.

ln 325: I suppose the authors mean "validated"? However, "identified" or "confirmed" is probably more appropriate?

Methods

ln 350: Need a new line for section heading.

ln 354: Remove "however". Replace "the interviews of the participants, who signed the written consent" with "interviews, and participants signed written consent".

ln 359-361: Give taxa names in full here. I think also provide authorities and higher taxa. Ensure names used are currently valid (see abstract comments). The authors should clarify their reference for snail names and/or identification, especially if they disagree with MolluscaBase.

ln 362: It seems to me that the details for how snails were collected in 4.3 should go together with the list of snails collected in 4.2

ln 370: Replace "the lightsource" with "a lightsource".

ln 374: Should not be italicised and should be lowercase, furcocercous is a term referring to a cercarial morphotype, not a species name.

ln 377: Does "released" mean the authors are referring here to only cercariae that emerged naturally vs from crushed snails?

ln 378: Comma for numbers >=1,000? Replace "for 3" with "three".

ln 284: Primer names and references?

ln 390: What is meant by "confirm" here? Identify? Delineate?

ln 392-393: References for primers?

ln 398-402: This sentence is too long.

ln 405: Use en-dashes, not hyphens. Remove extra whitespace after dashes.

ln 409-411: It is important to provide references for the sequence data used in alignments, to credit others whose previous work enables the current study.

ln 411: CLUSTAL vs CLASTAL?

ln 415: Comma for 1,000?

ln 417: rRNA

ln 418: Insert "as" after "used".

ln 422: Insert "the" before "purpose".

ln 433-436: Rearrange to: "Since most of the participants of the study were engaged in multiple professions, we built a Venn diagram using a Bio-Venn software to demonstrate the distribution of condition in different profession...". Spelling of "built".

ln 438: What were these analyses testing??

Conclusion

ln 441: Use "This study" or "We" vs "The study". "as" is italicised. Use "act" present tense, not "acted". Inappropriate use of "such as", use "specifically" or "namely" instead. Rearrange the sentence as follows: "This study confirmed that several FWS are intermediate hosts of schistosomatids in Bangladesh, specifically: L. auricularia, L. luteola, I. exustus, P. acuta and V. vivipara. The lymnaeid snails appear to be the most important hosts." Spelling of "lymnaeid".

ln 445: Remove "however". (Don't use both although and however in the same sentence like this).

Figure 1.

I do not think this figure is needed - it might be useful as a graphical abstract, but it serves no purpose in the manuscript. Figure A is just photos of the snail species examined. The graph in B reports simple numbers, i.e. prevalence per species - this could easily be provided in text or in a simple table. Figure C is a simple diagram of a cercaria. Most importantly, it is unusual and inappropriate to provide Methods within a figure caption - this information should instead occur in the Methods section of the manuscript.

Figure 2.

Remove methods from caption. How where the p-values calculated? There is no description of the statistical analyses in text.

Figure 3.

Again, the caption is mostly methods text and inappropriate. Three analyses are listed, and two gene regions, giving six potential phylograms - which are the two pictured? What is shown in A and B? The figure (in particular A) could be improved by indicating which are human vs livestock etc infecting species on the tree.

Figure 4.

Again, methods in caption. Caption needs to describe what is pictured, not how the data were collected. What is shown in each of A, B and C? Fig 4A adds is completely unnecessary. The percentage infected can easily be provided in text.

Figure 5.

"Mammalian" is spelled incorrectly.

Figure 6.

Again, the phrasing for the caption is largely inappropriate and confuses Methods text with caption text.

Reviewer 2 Report

This is a very interesting study, and I am impressed with the sample sizes.  I think that your study would benefit from increasing some explanation about some of your methodologies and also thinking about the language of both parasitology and ecology.  Some of the terms you used, I would use differently.  I would encourage you to be careful with how you use terms like "vector" and "niche" specifically.  Here are some comments I collected as I read your paper:

·      Title Comma should follow “vectors” not “and”

Abstract

·      31:  “equally distributed in all profession.”  Should be “equally distributed across all professions.”

Introduction

·      44-45: “more debilitating than leishmaniasis…”. What kind of leishmaniasis?  Make it clear that you are determining this based on DALYs.  Not sure why it even needs to be mentioned because it isn’t adding anything (at this point). It also follows a statement about helminths, and leishmaniasis is a protist, not a helminth.

·      58-60: “In addition, cercariae of some non-human schistosomes especially that of avian schistosomes…” Confusing.  Are you talking about species of schistosomes that include humans as the definitive host in the lifecycle or are you talking about humans as an accidental host?  The structure of the sentence could use revision and this would help with the clarity.

·      64 – “Although, the vertebrate hosts of schistosomes are wide, however, they use a few species…”. This is confusing .   Ithink you are trying to say that there are many vertebrate hosts for schistosoma spp., but there are fewer intermediate snail hosts for the parasite.

·      71 – “inflicting” do you mean “infecting”?

Results:

·      79-82: how many snails of each species?

·      Methods in the results section

·      The organization of information in this section makes it difficult to follow.

·      84-87 – it would be important to know sample sizes for each species ( I see them in the figure, but adding them to the text would be helpful).

·      100-101 – is vector the correct term here?  I would use intermediate host.  

·      109 “very small snails”… give the size range.

·      132-134 – the use of the term “niche” typically is referring to a specific role of an organism within a habitat, like snails using bottom sediments vs. snails using emergent vegetation.  I believe you are describing “habitats” rather than “niches” when you compare rivers, canals and beels.  A niche would distinct locations within a river, for example.  Not the river itself.

·      189-190 – confusing sentence structure.  ”Since”, “therefore” not used correctly. Just delete “therefore”.

·      200 – singular/plural lack of agreement

·      284 – “Snails…highest infection.”  What is meant by this?  More snails were infected? Intensity of infection was highest?  This is an important distinction.  I don’t believe you can talk about intensity of infections, so you should consider re-wording.

·      287-295- I think you need to make a stronger connection between the importance of buffalo and waterfowl to transmission of the Schistosomes. 

·      293-294- you switch from talking about definitive hosts (buffalo and waterfowl) to light – is there a connection? Explain – maybe this should be a new paragraph?

·      295- “…exposure of light to the infected snails is commonly used method to harvest cercariae [2]. Therefore, it can be predicted that infected snails migrated to the sunny area to be exposed to sunlight and release cercaria.”  This is an interesting assertion, but can you say this?  I think you need to test this – or provide context.  Were you measuring light intensity where you sampled?  Do you know that the positive snails came from places with higher light intensity? 

·      329 – how did you rule out other causes of urticaria?  How are you certain that the urticaria described was from cercaria?

·      367 – were equal efforts in sampling used from every location?  Did you use timed sampling, or did you sample until you removed a certain number of snails?  This is important information to include.

Reviewer 3 Report

The article „Mammalian and Avian Larval Schistosomes in Bangladesh: Molecular Characterization, Epidemiology, Molluscan Vectors and, Occurrence of Human Cercarial Dermatitis” describes results from the study on snails for identification of different species of schistosomes.

In my opinion the authors perform interesting results which can useful for parasitologist and also general readers.

However I have few issues for the authors which should be consider:

1.        please add Sanger before seqencing

2.       did the authors wonder why there are more parasites in small snails?

If yes please include paragraph about it in discussion part.

3.       The authors should include in disussion results from similar studies and compare results.

Round 2

Reviewer 1 Report

The authors have made a significant effort to respond to criticism and the manuscript has been much improved. I am now satisfied that all my more major original concerns have been addressed. However, many minor errors, inconsistencies and style errors remain present throughout, many of which were brought to the attention of the authors previously. 

Snail identification.

I am now satisfied that the authors have sufficiently determined the appropriate names to use for snail species.

Figures

The authors are justified in included all figures. The captions have been improved, but further improvements are needed (see end of document).

GenBank citations

I still maintain that the authors should provide references for the GenBank data used in their phylograms, to give due credit to the work of others. The authors response is that the accession number identifier is appropriate credit. I disagree. Citations provide the proper and full credit and are far more valuable to authors than merely listing accession numbers. The authors are making use of the data generated by others, they should acknowledge that contribution with full citation.

Writing errors

The grammar and English have been improved. However, throughout the manuscript, there remain many basic science writing errors, which I noted in the first review but have not been corrected. Specifically:

Hyphens: Hyphens should be used only to hyphenate words. When presenting a range (e.g. 1-10), do not use the hyphen symbol, use the en-dash symbol instead. To be clear, the issue here is not with hyphens introduced where words run over two lines. The issue is with misuse of hyphens other than to hyphenate, specifically when giving ranges.

Scientific names: When abbreviating a genus as part of a scientific name, the period should not be italicised. Please correct throughout.

Lists: When listing taxa, list in alphabetical order, unless there is a logical reason otherwise. Most lists in this manuscript are unordered.

Family names: Taxa family-rank names, like Schistosomatidae, must be capitalised. However, do not capitalise schistosomatid or schistosomatids. Check throughout.

Snail size.

The authors claim that infection prevalence was positively correlated with snail size. In the first review, I expressed concern that this claim was unsubstantiated because no data or analyses were provided. The response of the authors acknowledge that they do not have any critical analyses, but merely noticed "that lymnaeid snails <7 mm and Planorbid snails <5 mm of size did not carry any SC" and that larger snails seemed to be more frequently infected. Ok, fine - I have no problem with the authors providing their observations and speculations. However, they must make this clear; this is still presented as fact in the revised manuscript. The authors state: "Infection rates were higher in bigger snails". They cannot say this, because they have not demonstrated it. Please change to something like: "We observed that bigger snails were seemingly more frequently infected." - this makes clear that the observation is not a statistically supported claim. However, I still do not understand why the authors do not simply do the statistical work necessary to test this hypothesis. In the results section, they now provide size data for the snails, so, seemingly, they have all the data necessary to test whether size does indeed correlate with infections. In the response to the first review, the authors write: "In near future, we have a plan to reveal the matter by a controlled study." I do not understand why a controlled study would be required - the point of interest here is to report on the naturally occurring prevalence on SC.

Survey

In the methods, the survey is described as "cross-sectional" which I presumed to mean representative of the broader population. However, in the results, the authors indicate that only farmers, fishers and those who bathe/swim in natural water bodies were surveyed. So, did the survey find that 53.5% of the broader local population has HCD, or that 53.5% of farmers+fishers+bathers had HCD? The distinction here is likely sizable and significant. Also, it seems the sample size is now given only in the figure caption, nowhere in the manuscript text?

"Vector"

I still encourage the authors to avoid the use of "vector" - but ultimately I respect their choice of words.

Specific corrections and queries:

Abstract

ln 16-17: I am sorry, but I still take issue with the two opening sentences: "Schistosomiasis is a neglected tropical disease (NTD) caused by blood flukes (Schistosoma spp.). It affects a wide array of vertebrate hosts, including humans." The use of "It" (or in the previous version "The disease") referring to schistosomiasis is singular, but the authors are describing disease in a breadth of vertebrates caused by a breadth of schistosomatids. In this context, schistosomiasis is not one disease, but a family of diseases, plural. Likewise, in the first sentence, the authors state that schistosomiasis is an NTD, but in the second sentence state that schistosomiasis affects various vertebrates including humans - it is not an NTD in other vertebrates, NTD only applies to humans. Finally, I suspect the authors are including birds in "vertebrates" here? But in the first sentence they restrict schistosomiasis to be caused by Schistosoma spp. I suggest change to: "Schistosomiasis is a neglected tropical disease (NTD) caused by blood flukes (Schistosoma spp.). Schistosomatids affect a wide array of vertebrate hosts, including humans." Or "Schistosomatids, the blood flukes, exploit a variety of vertebrates, including humans and causing schistosomiasis, a neglected tropical disease".

ln 18: Remove "different species of". Perhaps replace with "multiple".

ln 18: Schistosomatids should be lowercase.

ln 18: Remove the second occurrence of schistosomatid here.

ln 21: The period in "L. luteola" should not be italicised.

ln 23: Please ensure periods are not italicised.

ln 24: Please ensure periods are not italicised.

ln 25: Please ensure periods are not italicised.

ln 25-26: Replace "The schistosomatids infection rate in snails..." with "Prevalence in snails..."

ln 27-28: Remove "in Bangladesh" - implicit here.

ln 28: Schistosomatids needs to be lowercase.

ln 29: Please ensure periods are not italicised.

ln 30: To me, "cross-sectional" implies representative of the broader population, but in the results section the authors write: "we surveyed HCD in the study areas by interviewing individuals engaged in farming and/or fishing as an occupation or those who used natural water bodies for swimming or bathing". That implies the survey was not cross-sectional but targeted at risk groups? 53% infection in farmers/fishers is different to 53% infection in the general population.

ln 32: Seems to me that by "all professions" there were actually mainly just two, farmers and fishers?

ln 33: Schistosoma needs to be italicised. Do the authors mean shistosomatids here though - i.e. do they intentionally exclude T. szidati here?

Introduction

ln 38: "Schistosomiasis" should not be italicised.

ln 41: Whitespace needed before "[1]".

ln 44: It is necessary to add "in humans" after "helminth infection", because the subject in the previous sentence was schistosomatids in livestock, whereas here it is humans.

ln 47: The Oxford comma after "Africa" is not necessary and the sentence would read more easily without it. 

ln 48: The commas after "japonicum" and "haematobium" should not be italicised. The periods in "S. haematobium" and "S. mansoni"  should not be italicised. 

ln 48: Why are the species not listed alphabetically?

ln 48-49: The authors contradict themselves. In the first clause, S. japonicum is a primarily a human parasite, whereas in the second clause it is zoonotic, implying it is not primarily a human parasite.

ln 49: Replace "while" with "whereas" - the latter is for comparisons, the former invokes time, i.e. A while B.

ln 49-50: The periods in "S. japonicum", "S. spindale", and "S. bovis" should not be italicised.

ln 51: "An adult schistosomatid" or "Adult schistosomatids" but not "An adult schistosomatids".

ln 51: Replace the hyphen in "3-10" with an en-dash. Hyphens should be used only to hyphenate, use en-dash for ranges.

ln 52: Use "lay" vs "lays" because "may... lay" vs "may... lays".

ln 53: I though "where" was better than "at which" here.

ln 54: Remove "up" - they do not pair "up" or "upwards", they simply pair/partner.

ln 54: The comma after "indicum" should not be italicised.

ln 54-56: The periods in abbreviated genera should not be italicised.

ln 56: List species alphabetically - i.e. place Schistosoma before Trichobilharzia.

ln 58: Should "granuloma" be plural?

ln 60: "Severe" is used twice in this sentence. I suggest remove this first occurrence.

ln 61: "Swimmer'" should not be capitalised as it is not a proper noun.

ln 63: "Schistosomatids" should not be capitalised.

ln 66: Is "Schistosoma" correct here? Or is "schistosomatids" better? I.e. are the authors intending to restrict to non-avian parasites here?

ln 67: Surely the authors do not mean "[there are] very few freshwater snails"? Certainly there are far fewer freshwater snails than marine, but there is still reasonable freshwater richness and certainly far more species of freshwater snail than species of Schistosoma. Perhaps the authors mean few in Bangladesh specifically? Even then, I suspect the statement is exaggerated? Or perhaps the authors mean there are few species of snails which are used by schistosomatids or species of Schistosoma? In any case, at least remove "very" - it serves only to suggest the statement is exaggerated. The sentence is also grammatically incorrect. Should the comma after "snails" be a period? The abbreviation FWS is not introduced. Personally, I think it would be simpler to use "snail" vs "FWS" throughout the manuscript (it being implicit that they are freshwater hereafter). 

I suppose the intended meaning is that fewer species of snails are used than vertebrate hosts, or perhaps the taxonomic breadth of snails used is narrower than vertebrates? Needs to be clearer.

ln 69-71: I find this statement inadequate - the utility of PCR was evaluated... and?? Was it found to be suitable for diagnosis or not? What point are the authors attempting to make by including this sentence?

ln 71: "Schistosomatids" should not be capitalised.

ln 71: Remove "different". Different to what? The meaning here is "several" or "multiple"?

ln 73: I think "described" is the wrong word here - "identified"? But I take some issue with this because the authors (still) do not clearly express which species they found in which snails.

ln 74: I suggest remove "key".

ln 74: Remove "with the parasite", it is implicit because of "infections".

Methods

ln 79: "Ethical Committee" should not be capitalised. Besides, this should be "ethics" committee.

ln 80: The study did not involve human samples, but it also did not involve dealing with vertebrates. I think the authors might as well expand and make this clear here: "We did not handle any human nor vertebrate samples/tissues."

ln 83: "Collection" should not be capitalised.

ln 85: The Oxford comma is unnecessary here and should be removed.

ln 87-90: I still recommend the authors indicate higher taxa for snails here.

ln 87: I think the correct authority for Lymnaea luteola is "Lamarck, 1822", not "(Lamarck, 1822)", the parentheses matter!

ln 88: I think the correct authority for Physa acuta is "Draparnaud, 1805" vs "(Draparnaud, 1805)".

ln 89: Whitespace missing after "Viviparus bengalensis".

ln 89: I think the authority for Brotia is "Adams, 1866" vs "(Adams, 1866)". Also, the authority is for the genus and so should be given as "Brotia Adams, 1866 spp." vs "Brotia spp. Adams, 1866".

ln 90: As for Brotia above, Thiaria has no parentheses on authority, and authority should come before the "spp.".

ln 91: I think replace "different" with "various".

ln 91: I think probably just "scoop" vs "scooped".

ln 93: It seems to me that "i.e." might be more appropriate than "e.g." here?

ln 94: Habitat within the habitat? Perhaps micro-habitat would work here?

ln 101: Simplify "Each of the snails was kept in a separate glass test tube" to "Snails were kept in separate glass test tubes".

ln 103: Insert comma after "3 h", and replace "following" with "as per".

ln 106: Here the authors use "furcocercous" without providing any definition or description for the benefit of the reader. That detail comes later in the results at ln 185-188 and also in Fig. 2. This information should instead be placed here, at the first mention of furcocercous", with reference to the figure.

ln 110: Replace "released" with "recovered".

ln 116: What was the purpose of the 28S PCR? I presume that because cox1 was used to identify only bird schistosomes below, 28S was used to identify mammal schistosomes? This is not clear.

ln 117: These primers do not have names?

ln 123: I think by "confirm" the authors mean "identify"? I.e. the purpose of using cox1 was to identify species.

ln 124: Remove "cox1:" within the parentheses.

ln 133: Here "with the same primers for 28S rRNA and the COX1 gene" is confusing. I presume the authors do not mean that one set of primers was used for both 28S and cox1, but rather that the amplification primers for each gene were also used for sequencing. But this is not true? In the previous section the authors state different or at least additional primers were used for sequencing vs amplification of cox1.

ln 133: Be consistent - why is cox1 given as COX1 here?

ln 133: "A sequencing primer was used as described above..." pertains only to cox1, correct? This is confusingly present and not clear. Why are sequencing primers including both in this section and the previous section?

ln 138: Please replace hyphens with en-dashes. Please only use hyphens to hyphenate!

ln 146-149: I suggest rephrase as follows:

"Phylogenetic trees were constructed using maximum likelihood (ML), neighbour joining (NJ), and minimum parsimony (MP) methods in MEGA X [23], based on the Tamura-Nei model with bootstrapping (1,000 replicates) and otherwise default settings."

ln 150: Correct "28r RNA" to "28S rRNA", and "COX1" to "cox1".

ln 151: Simplify "Chemaerohemecus trondheinensis, a trematode of the family Strigeidae" to "Chemaerohemecus trondheinensis, (Trematoda: Strigeidae)".

ln 161-162: This part about vertebrate animals would be better placed in the ethics section above vs the survey section here.

ln 166-167: Although ultimately harmless, this sentence adds nothing: "Information was recorded and data were analyzed." That is implicit but also vague and so adds nothing.

ln 173-176: The authors do not make clear in this section which data these analyses were performed on. Presumably the survey data from the section above? Although the sample size was <30, whereas the survey above included 400 participants? But which questions were specifically tested with z-tests? What was the purpose of the analyses? Here, the authors are simply stating "we did analyses", but analyses of what and why?

Results

ln 179: "Schistosomatids" should be lowercase.

ln 180-182: Delete the first sentence, it is repeating methods. Also, from the second sentence, delete "Among the samples tested", just start the paragraph with "Schistosomatid cercariae...".

ln 183-184: Please list the species alphabetically. Please do not italicise periods.

ln 185: "schistosomatids" plural.

ln 185-188: This detail for identifying SC belongs earlier, where "furcocercous" is first used. I.e. these few lines are methods, not results, because they explain how the authors identified SC. Also "initial experiment" is confusing - what initial experiment? And is experiment the right word? Perhaps survey?

ln 197-199: There are problems with this sentence. Use "suggest" instead of "suggested". Although results should be presented past tense, any inference from those results should be present tense. So here "obtained" remains past tense, but "suggest" in present tense, because the obtained results still suggest (at present vs in the past only) whatever conclusion is made. But in any case, do not use "suggest" at all, because the statement here is that the finding was statistically significant, that is a fact, no "suggestion" about it. Do not start with "The results obtained suggest...", just state the finding. Need "in" or "for" before "L. luteola". The statement "SC was significantly higher (p<0.05) [in] L. luteola" is a comparison, but it is not made clear what L. luteola is being compared to. Finally, please ensure the period is not italicised in each of the three species names. Finally, I will stress again, multiple species of schistosomatid are involved here, but the authors are lumping prevalence together - that is fine, but it needs to be made clear. I suggest rephrase this sentence as follows: "The infection rate with SC, pool for all schistosomatid species involved, was significantly higher for L. luteola (11.1%) than for all other snails, and for L. auricularia (5.3%) versus all other snails except L. luteola. The lowest prevalence occurred in V. bengalensis (0.4%) (Figure 2B)." 

ln 200: Whitespace missing between "bigger" and "snails". Although perhaps "larger" is better - somehow seems a bit more formal. Can remove "than is smaller snails" - this is implicit. No p-val is provided to support this. I understand that the authors did not actually confirm or test this statistically? That is fine, I have no problem with the authors reporting there observations, but they must make clear that these are observations only. The authors cannot state "Infection rates were higher..." because they have not demonstrated this. Instead, phrase something like: "We observed that infection rates were seemingly higher....".

ln 200: Cannot start a sentence with an abbreviation. Give "Lymnaea" in full.

ln 201: I am confused by the numbers here. According to the authors, L. auricularia greater that 16-31 (20 ± 5.5) mm had a higher rate of infection? Higher than what, snails less than 16 mm? Then why not phrase as L. auricularia greater than 16 mm had a higher rate of infection than individuals smaller than 16 mm? Or is 16-31 (20 ± 5.5) mm the range of L. auricularia examined?

ln 201: The numbers provided here indicate that the authors did record snail size. In that case, I do not understand what is preventing a simple statistical test of size vs infection prevalence, to test their observation that larger snails were seemingly more frequently infected. It seems to me they have all the data necessary?

ln 201: The periods in "L. luteol" and "I. exustus" should not be italicised.

ln 201: Remove trailing zero from "6.250".

ln 201-202: Replace the hyphens with en-dashes in "16-31", "9-16" and "7-18".

ln 202-203: The following sentence is a repeat of previous sentences: "The rate of infections increased with the size of snails."

ln 221: Remove "Different". "Schistosomatids" needs to be lowercase.

ln 226-227: I suggest rephrase this sentence: "We also collected snails with a distinct record of exposure to sunlight or not." to "We recorded sunlight exposure when collecting snails".

ln 227: Delete "The results obtained revealed that", start the sentence at "Snails...".

ln 228-229: Again, as with snail size, it seems the authors have all the data to perform the necessary simple statistical analysis to test their hypothesis. I appreciate that categorising snails as "in shade" or "in sun" is crude - but the authors are seemingly convinced of an effect, why not test it and add a p-value?

ln 231: Remove "different". "Schistosomatids" needs to be lowercase.

ln 233: Remove "different".

ln 235: Should this "e.g." be "i.e."? Or perhaps just remove the "e.g." altogether?

ln 236: Replace the hyphens with en-dashes at "ON597438-ON597447", "ON597471-ON597474" and "99.41-100%".

ln 236: Hm... "were identical (99.41-100% identities)". 99.41% is not quite "identical". Perhaps "essentially identical" or "mostly identical" or "nearly identical".

ln 237: Please ensure periods are not italicised for "S. indicum", "S. spindale", "S. nasale", "S. incognitum" and "T.".

ln 240: "showed Single Nucleotide Polymorphism (SNP) in a very few (1-3) points" with "revealed few (1-3) single nucleotide polymorphisms (SNPs)"

ln 241-242: I think rephrase this "However, we our sequences did not match with sequences specific for..." as "Our sequencing effort did not detect any matches with...".

ln 242: Why are the species not presented in alphabetical order? Please ensure periods are not italicised: "S. mansoni, S. japonicum, and S. haematobium".

ln 243: Replace "particularly" with "at least".

ln 244: The sequences are rDNA not rRNA (the gene/product is rRNA). So either "28S rDNA" or "28S rRNA gene" but not "28S rRNA".

ln 245: Either remove "of our isolates" or replace "of" with "from".

ln 246: DO not italicise "28Ss".

ln 246: Same again, the sequence data are DNA, not RNA, although they are sequences of RNA products.

Why are the species not presented in alphabetical order? Please ensure periods are not italicised: "S. indicum, S. spindale, S. nasale, S. incognitum, S. mansoni, S. japonicum, and S. haematobium"

ln 247: Replace hyphens with en-dashes.

ln 248: Replace hyphen with en-dash.

ln 249-250: The accession numbers are not sufficient - the authors need to cite sources. The authors are using the data of others, cite appropriately to give the recognition deserved.

ln 250: Add ", respectively," after "JQ408705)".

ln 253: The abbreviation "cox1" was already established in methods, do not provide in full here.

ln 253-255: Please do not italicise periods in species names. Why are the names not ordered alphabetically?

ln 256: Replace hyphen with en-dash.

ln 256: The period in "T. szidati" should not be italicised.

ln 258-261: I do not think this sentence is needed. Harmless I suppose - although in Fig4 T. ocellata clusters among T. szidati so can hardly be described as "distantly related". Please ensure periods are not italicised.

ln 262: The period in "T. szidati" should not be italicised. Please correct both occurences on this line.

ln 262: "By molecular analysis, we revealed that SC of T. szidati were more common in P. acuata." More common that what? Which other snails had this schistosomatid species? Quantify the statement.

ln 263: Swap "only harbored" to "habored only"

ln 264: Delete "caused infection".

ln 264-265: "...mainly infected lymnaeid snails planorbid snails."?? I suspect it should be only lymnaeid or planorbid, not both? Why not be more specific, which species of schistosomatid were detected in which snail species?

ln 265: "None of the species were isolated from V. bengalensis." But in the previous statement T. szidati infects V. bengalensis?

ln 270: Delete "the different" and "in the study areas".

ln 271: The period in "T. szidati" should not be italicised.

ln 272-273: In the methods, this survey was described as cross-sectional. Here, the authors indicate that only farmers, fishers and those who bathe/swim in natural water bodies were surveyed. So, did the survey find that 53.5% of the population has HCD, or that 53.5% of farmers+fishers+bathers had HCD?

ln 280: Insert "had" before "become".

ln 283: Replace "were" with "are" and "penetrated" with "penetrate".

Discussion

ln 298: The authors cannot use "it" like this to mean "Schistosoma spp." because "it" is singular whereas the latter is plural. I still do not really understand the term "eco-health" - are not all parasites "eco-health" issues? I suppose the point being made here is because of the complex life-cycle, management of schistosomiasis and HCD in humans requires intersection with ecological study and management?

ln 299-300: Not true. There were only four species of Schistosoma plus one species of Trichobilharzia - the subject in the previous sentence is "Schistosoma spp.", so the authors cannot then say in this sentence that they found five species, they need to specify five species of schistosomatids.

ln 300: Replace "verified the different" with "identified".

ln 302: Replace "presented" with "documented".

ln 304: List the taxa alphabetically! Please ensure periods are not italicised.

ln 305: Schistosomatids should be lowercase.

ln 306: Please ensure periods are not italicised.

ln 308: Schistosomatids should be lowercase.

ln 310: Replace "vital" with "substantial". This snail clearly isn't "vital", just the most important, as the authors showed other snails also acted as hosts.

ln 311: Please ensure periods are not italicised.

ln 312: The "other species" here excludes L. auricularia and L. luteola? That is not quite clear.

ln 313: Schistosomatids should be lowercase.

ln 313: Cannot start a sentence with an abbreviation. Give Indoplanorbis in full.

ln 315: Please ensure periods are not italicised.

ln 320: I think replace "SC" with "avian SC" here?

ln 320: Please ensure periods are not italicised.

ln 325-326: Replace "is currently" with "was previously".

ln 326: Please ensure periods are not italicised.

ln 327: Please ensure periods are not italicised.

ln 354: Insert "water" after "low lying"?

ln 356: This number is for domestic ducks or all ducks?

ln 359: Replace "by" with "with". Insert "the" before "main".

ln 360: Schistosomatids should be lowercase.

ln 360: Do ducks "act as [the] main source of contaminations" only for avian schistosomatids? The authors have "particularly" avian schistosomatids, but is it not "only" avian schistosomatids? I presume ducks are only carrying and spreading avian schistosomatids, not bovine ones too?

ln 367: Replace "confirmed" with "identified" or "detected".

ln 367-368: Please alphabetise list. Please ensure periods are not italicised.

ln 368: The current phrasing suggests a single primer set was used to generate both 28S and cox1. Replace "a primer set" with "primer sets".

ln 369: Replace "COX1" with "cox1" and italicise.

ln 368-370: This sentence is not needed.

ln 374: I don't understand: the authors "plan" to sequence amplicons, but in the very next sentence "Our newly recovered sequences...". The authors have already produced sequences! The purpose here (at least it seemed to me) was to use sequencing to identify the schistosomatid species involved - and the authors have done that. I understand they may wish to do further sequencing in the future to answer further questions, but I see no reason to mention that here. The way it is currently presented is confusing - it isn't clear what the authors plans are or why, why is more sequencing required? What bioinformatic analyses - what questions would these answer? I think better to just focus on what was found here.

ln 375: Delete extra whitespace in accession numbers. Replace hyphen with en-dash.

ln 376: Replace hyphen with en-dash.

ln 376-377: Please alphabetise list. Please ensure periods are not italicised.

ln 378-379: Please ensure periods are not italicised.

ln 380: Please ensure periods are not italicised.

ln 383: Please ensure periods are not italicised.

ln 385: The authors use "extensive" here. I don't think the number of extraction nor sequences was given - I counted about 14 sequences generated? Is that extensive? Certainly the snail and schistosomatid collecting effort was extensive, but the sequencing effort?

ln 386: Italicise "S. mansoni" [except the period!].

ln 386: Please ensure periods are not italicised.

ln 387: Please ensure periods are not italicised.

ln 390: Please ensure periods are not italicised.

ln 390-391: This is confusing - the authors seemingly acknowledge T. ocellatus is a juior synonym but continue to use it as if it were a valid name. Replace "T. ocellatus" with "T. szidati (as its junior synonym T. ocellatus, see [37])" and delete ", which at present is synonymous with T. szidati [37]."

ln 392: Remove "very".

ln 401: By "multiple professions" here and "irrespective of profession" elsewhere, the authors are referring just to fishers and farmers?

ln 405-406: It seems that rather than a "cross-sectional" survey the authors actually mostly surveyed farmers and fishers. So really the "high prevalence and indiscriminate distribution" of HCD here is high and indiscriminate among the groups mostly likely to be affected. Statements alluding to "indiscriminate" "irrespective of profession" are a bit misleading when the authors only survey the small number of professions (fishers and farmers) mostly likely to be affected. I think the authors should change the broad point they are making here. Rather than claiming they have found evidence of high and indiscriminate prevalence within Bangladeshi, they should emphasise that they surveyed the communities most likely to be affected and found high prevalence within these at-risk groups.

ln 410: Either use superscript for "3rd", or, better, use "third" and "forth".

Conclusion

ln 417: Why be vague here? Replace "several" with "five" and remove "such as".

ln 420: The authors should be careful with "strongly" - this implies they have "strong" evidence, which, for size and sunlight, they do not (no p-values were given).

ln 421: Replace "recovery" with prevalence". The mentioned factors explain and influence prevalence. Recovery is dependent on methods and technique.

ln 423: Well sure "particularly villagers" but the survey suggests "especially farmers and fishers".

References

ln 453: "Schistosoma mansoni" needs to be italicised. "mansoni" needs to be lowercase.

ln 473: "Schistosoma mansoni" needs to be italicised. "mansoni" needs to be lowercase.

ln 476: "Is" should be lowercase.

ln 476: "Biomphalaria glabrata" needs to be italicised. "glabrata" needs to be lowercase.

ln 477: "Schistosoma mansoni" needs to be italicised. "mansoni" needs to be lowercase.

ln 483: Why are authors in all caps?

ln 488: "Schistosoma japonicum" needs to be italicised. "japonicum" needs to be lowercase.

ln 490: "Schistosoma mansoni" needs to be italicised. "mansoni" needs to be lowercase.

ln 498: "Trichobilharzia" needs to be italicised.

ln 507: Opening " symbol has no matching closing symbol.

ln 514: "Physa marmorata" needs to be italicised. "marmorata" needs to be lowercase.

ln 515: "Trichobilharzia" needs to be italicised.

ln 530: "Schistosoma indicum" needs to be italicised. "indicum" needs to be lowercase.

ln 531: "Schistosoma spindale" needs to be italicised. "spindale" needs to be lowercase.

ln 532-533: "Schistosoma nasale" needs to be italicised. "nasale" needs to be lowercase.

ln 534: "Schistosoma incognitum" needs to be italicised. "incognitum" needs to be lowercase.

Figures

Fig. 1. Caption: Remove "Map showing", just "Study areas" will suffice, although I think "area" singular. "Map" is not needed in the caption, the figure is clearly a map! I think "engine" should be lowercase. I suggest rephrase caption as follows: "Study area. Prepared with Landsat satellite images and the Google Earth engine, see [19] for detailed method."

 Fig. 3. Caption: Replace "A similar number of snails was collected" with "A similar number of snails were collected", and replace "A comparable number of snails was collected" with "A comparable number of snails were collected". Use a comma for numbers >= 1,000.

Fig. 4. Caption: The authors need to state from which sort of analyses these trees were constructed. Remove italics for "28S rRNA". The statement "showing different species of Schistosomatids affecting livestock" is not true, human schistosomatids are included too. "Schistosomatids" should be lowercase. Replace "COX1" with "cox1".

Reviewer 2 Report

Thank you for your attention to all of the comments.  This is a much stronger paper as a result.  My only comment at this point is that you interchange "vector" and "intermediate host".  I suggested "intermediate host", but did look through the literature and see that either would be appropriate.  However, you should choose one.  It would probably be easier to use "vector".  The intermediate host references happen in the discussion and conclusions.

Author Response

Reviewer 2

Thank you for your attention to all of the comments.  This is a much stronger paper as a result.  My only comment at this point is that you interchange "vector" and "intermediate host".  I suggested "intermediate host", but did look through the literature and see that either would be appropriate.  However, you should choose one.  It would probably be easier to use "vector".  The intermediate host references happen in the discussion and conclusions.

Response: Thank you so much for appreciating our efforts. We have replaced the term intermediate hosts with ‘vector.